# AI outperforms humans in establishing interpersonal closeness in emotionally engaging interactions, but only when labelled as human
Tobias Kleinert [1] ✉, Marie Waldschütz [1], Julian Blau [1], Markus Heinrichs[1] ✉ & Bastian Schiller [1,2] ✉

With the increasing accessibility of large language models to the public, questions arise about whether, and under what conditions, social-emotional interactions with artificial intelligence (AI) can lead to human-like relationship building. Across two double-blind randomised controlled studies with pre-registered analyses, 492 participants engaged in dyadic online interactions using a modified, text-based version of the 'Fast Friends Procedure' (a method designed to enable rapid relationship building), with pre-generated responses by either human partners or a minimally prompted large language model. When labelled as human, the AI outperformed human partners in establishing feelings of closeness during emotionally engaging 'deep-talk' interactions. This striking effect appears to stem from the AI's higher levels of self-disclosure, which in turn enhanced participants' perceptions of closeness. Labelling the partner as an AI reduced, but did not eliminate, relationship building, likely due to participants' lower motivation to engage in interactions with an AI, reflected in both shorter responses and reduced feelings of closeness. These findings highlight AI's potential to relieve overburdened social fields while underscoring the urgent need for ethical safeguards to prevent its misuse in fostering deceptive social connections.

Fuelled by technological breakthroughs in neural network modelling and the rapidly advancing development and accessibility of large language models (LLMs), our experiences and evaluations of social interactions with artificial intelligence (AI) have fundamentally changed in recent years[1–6]. Indeed, growing evidence suggests that LLM-generated content can facilitate communication that not only feels similar to interaction with human agents but, in certain aspects and contexts, may even surpass it. For example, participants tend to evaluate LLM-generated content as more empathic than human-generated content, particularly when they are unaware of its non-human origin[7–10]. This has motivated research on the potential translational value of such findings, reporting beneficial effects of LLM-generated communication in therapeutic contexts[11,12]. These findings show that LLMs can be used to generate human-like responses during social communication and first pioneering evidence also suggests that some degree of human-AI relationship formation is possible[13–15]. However, it remains an open question

whether, and under what conditions, humans build relationships with AI to the same extent as with other humans, especially in the early stages of building a new relationship to a previously unknown other. The present study aims to fill this research gap by investigating differences in relationship building between initial interactions with humans versus AI (i.e. LLM-generated content).

Driven by the rapid evolution of AI's communicative abilities, theoretical accounts of human-AI interaction are being refined. Early accounts explained social behaviour towards computer-mediated technology as the 'mindless' application of social heuristics (e.g. politeness, stereotyping) to encounters with machines that display interactive features, such as speech (e.g. Social Response Theory and the associated Computers-Are-Social-Actors paradigm[16,17]). With the emergence of more advanced AI technologies approaching human-level communication, existing theories of human interpersonal relationships (e.g. Social Penetration Theory and Social

[1]Department of Psychology, Laboratory for Biological Psychology, Clinical Psychology, and Psychotherapy, Albert-Ludwigs University of Freiburg, Freiburg, Germany. [2]Department of Psychology, Laboratory for Clinical Neuropsychology, Heidelberg University, Heidelberg, Germany. ✉e-mail: tobias.kleinert@psychologie.uni-freiburg.de; heinrichs@psychologie.uni-freiburg.de; bastian.schiller@psychologie.uni-heidelberg.de

Exchange Theory[18–20]) have been applied to human-AI interactions. However, AI still lacks certain human characteristics such as agency, autonomy, memory, and a personal history. Therefore, some have questioned the applicability of human relationship theories to human-AI interactions, emphasising the need for new theoretical frameworks grounded in empirical evidence on the nature of these relationships[14].

An argument frequently raised in public media discussions about the differences in communicative abilities between humans and AI is that AI-generated content may be at the level of human-generated content in some domains, but not in the 'uniquely human' domains such as emotion[21–24]. Fittingly, research shows that human interaction partners are preferred over AI ones, particularly in domains involving emotion[25,26]. These caveats contrast with evidence from the field of 'emotional AI', which suggest that LLMs possess a marked ability to recognise and respond appropriately to emotions[27–30]. Thus, when trying to understand the differences in relationship building between humans and AI, it is essential to consider the emotional intensity of the communication content. In other words, are there differences in AI's ability to engage in superficial small-talk versus deep-talk communication on more personally meaningful and emotionally charged topics that require a higher level of self-disclosure[31]?

Existing theories of human-AI interaction also fail to consider the impact of attitudes towards AI. As AI's social capabilities continue to evolve, concerns and reservations about interactive technological devices are growing in parallel. Indeed, many hold negative attitudes towards AI, perceiving it as unnatural and sometimes even threatening[32,33]. This scepticism is reflected in intense media debates about AI potentially replacing humans' unique socio-emotional and cognitive abilities[34], creating a paradox: Although communicating with AI can foster relationship building, once people realise they are interacting with AI, these effects seem to dissipate. For example, the perceived superiority of AI-generated responses over human-generated responses to the description of an emotional situation was reversed once participants realised that the response came from an AI[7]. When considering possible applications of AI in clinical and psychotherapeutic settings, such negative attitudes towards machine interaction become problematic, as the source of communication must be ethically revealed. Therefore, it is crucial to understand how labelling the communication source (as AI or human) influences relationship building.

The present pre-registered research, consisting of two double-blind randomised controlled studies, examines relationship building in social-emotional interactions with AI and humans (source human vs. source AI), and its modulation by the emotional intensity of the interaction (small-talk vs. deep-talk; study 1) and the labelling of the source of the content (label human vs. label AI; study 2; for details on specific hypotheses, see pre-registration available at https://osf.io/chdx7). Both studies were conducted online simultaneously and included some shared data to enhance comparability between studies, prevent participants from taking part in both studies, and avoid history effects, which are likely in a rapidly evolving field such as AI interactions. History effects are important to consider, as participants' attitudes towards, and interactions with AI can change significantly over time[35], potentially leading to observed differences in AI interactions that reflect broader societal shifts rather than effects of experimental conditions. To mimic everyday relationship building, 492 participants engaged in a 15-min online communication task (i.e. the Fast Friends Procedure, or FFP) designed to induce interpersonal closeness to an unfamiliar interaction partner through escalating mutual self-disclosure in a series of turn-taking questions[36–38]. We specifically selected the FFP because it was designed to enable the rapid development of interpersonal closeness in the early stages of relationship building between previously unacquainted partners[36], which was the focus of our study. Unbeknownst to participants, responses to FFP items were pre-generated either by a minimally prompted LLM or by human interaction partners who performed the FFP in a laboratory environment (for all items and responses, see Supplementary Table 1). We prompted the AI to respond from the perspective of fictional characters rather than in its original form to enable it to answer personal questions and keep basic character information (name, age, place of residence, field of study) consistent with human partners. Relationship building was operationalised by self-reports of perceived interpersonal closeness to the interaction partner. Furthermore, we applied exploratory automated linguistic analysis using the Linguistic Inquiry and Word Count system (LIWC[39,40]) to investigate whether any identified differences in relationship building between conditions could be explained by variations in self-disclosure levels of interaction partners (i.e. AI-generated characters or humans) and/or participants. In the LIWC, self-disclosure is operationalised as the number of words related to the self, emotions, and social processes[41]. We also analysed the response length of interaction partners and participants as a measure of social motivation, with longer responses indicating greater motivation.

In summary, the hypotheses we tested in study 1 were: Hypothesis 1: We expect that deep talk interactions with an AI will lead to a significant increase in interpersonal closeness compared to a baseline measure. This hypothesis is being tested to validate findings suggesting that relationship-building with AI is possible[7]. We additionally examine whether small-talk interactions will also generate increases in closeness. Hypothesis 2: We expect higher interpersonal closeness towards the interaction partner after interactions with humans compared to interactions with an AI across both small-talk and deep-talk interactions (main effect of the factor 'source identity'). This assumption builds on the traditional view that interactions with humans should elicit greater closeness than interactions with AI, as social interactions are fundamentally rooted in human behaviour[24]. Hypothesis 3: We expect higher interpersonal closeness towards the interaction partner after deep-talk interactions compared to small-talk interactions across interactions with humans and AI (main effect of the factor 'emotional intensity'). This hypothesis relies on the idea that self-disclosure on emotional topics is a key driver of early relationship-building[36]. Hypothesis 4: We expect that the differences (human > AI) regarding interpersonal closeness towards the interaction partner are larger after deep talk interactions compared to small talk interactions (interaction effect between the factors 'emotional intensity' and 'source identity'). This hypothesis assumes that AI may adequately mimic non-personal small-talk but not personal deep-talk, as emotions are widely considered a uniquely human domain[22], although more recent research (some published after our pre-registration) suggests otherwise[10].

The following hypotheses were tested in study 2: Hypothesis 5: We expect that deep talk interactions with an AI will lead to a significant increase in interpersonal closeness compared to a baseline measure, even if participants are informed that they will interact with an AI. Although research indicates that people often hold reservations about AI interactions[7], we anticipate that participants will still develop some degree of perceived closeness with the AI, reflecting the human tendency to respond to human-like artificial agents as if they were real social partners, a phenomenon known as anthropomorphism[16,17]. Hypothesis 6: Analogous to study 1, we expect higher interpersonal closeness towards the interaction partner after interactions with humans compared to interactions with an AI across both 'label human' and 'label AI' interactions (main effect of the factor 'source identity'). Hypothesis 7: We expect higher interpersonal closeness towards the interaction partner when participants are informed that they interact with a human compared to when participants are informed that they interact with an AI across actual interactions with humans and AI (main effect of the factor 'source label'). This hypothesis is based on findings indicating reservations towards social interactions with AI[7]. Hypothesis 8: We expect that the differences (human > AI) regarding interpersonal closeness towards the interaction partner are larger when participants are informed that they are interacting with a human compared to when they are informed that they are interacting with AI (interaction effect between the factors 'source identity' and 'source label'). This hypothesis draws on the assumption that the anti-AI bias in the AI-labelled condition would reduce feelings of closeness regardless of the actual source identity, whereas in the human-labelled condition, the expected advantage of genuine human responses would be more apparent.

## Methods

### Study design

Study 1 is a double-blind randomized controlled trial with a $2 \times 2$ between-subjects design including the factors 'source identity' (human interaction partner [hereafter referred to as 'source human'] vs. fictional characters created by a LLM [hereafter referred to as 'source AI']) and 'emotional intensity' (small-talk vs. deep-talk). Accordingly, participants were randomly (but evenly) distributed among the four treatment groups 'source human/small-talk', 'source human/deep-talk', 'source AI/small-talk', and 'source AI/deep-talk'. All participants of study 1 were informed they would interact with a human, which was a deception for the two treatment groups who actually interacted with an AI (for details on the debriefing, see Procedure/Online experiment). In study 2, we re-analysed selected data from study 1 along with other data collected for study 2. Specifically, study 2 was another double-blind randomised controlled trial with a $2 \times 2$ between-subjects design including the factors 'source identity' (source AI vs. source human) and 'source label' (label human vs. label AI). This study focused on deep-talk interactions only. Participants were randomly (but evenly) distributed among the four treatment groups 'source human/label human', 'source AI/label human' (both collected in study 1), 'source human/label AI', and 'source AI/label AI' (both collected in study 2). Accordingly, participants in the treatment group 'source human/label AI' were deceived about the identity of their interaction partner. In both studies, perceived interpersonal closeness to the interaction partner was used as the main dependent variable.

### Sample

We recruited 18- to 35-year-old heterosexual female and male university students for our study, focussing on friendly, non-romantic relationship building among individuals of the same self-reported gender. Gender was assessed using the following item: 'Please indicate your gender' (response options: 'female', 'male', 'diverse'). Participants were excluded if they had mental health challenges, were undergoing current psychotherapeutic, neurological, or psychiatric treatment, or were abusing alcohol or drugs (for details on exclusion criteria, see ref. [42]). Using G*Power[43], we performed an a priori power analysis for ANCOVA (Analysis of Covariance; fixed effects, main effects and interactions; numerator df: 2, number of groups: 4; number of covariates: 4; alpha = 0.05, power = 0.80, small to medium effect size $f = 0.175$; based on average effect sizes in social psychology and neuroscience[44,45]). This analysis yielded a required sample size of $n = 318$ for each study.

Expecting a drop-out rate of ~10%, we recruited a total sample of 359 participants for study 1. From this initial sample, 37 participants were excluded due to the following reasons: answering incorrectly to at least one out of two attention control items during the questionnaires ($n = 28$); an unrealistic total duration of the experiment of less than 50% of the expected minimum duration based on pre-tests (i.e. <20 min, $n = 2$); expressing doubts about the cover story of a live interaction in the experiment ($n = 6$); responding in a different language ($n = 1$). Thus, the final sample size analysed in study 1 was $n = 322$ (age: $M = 23.46$, $SD = 3.19$, range: 18–35; 168 female participants, 154 male participants). For study 2, we recruited an additional sample of 179 participants, out of which 9 participants were excluded due to either incorrectly answering to the attention control items ($n = 8$), or an unrealistic total duration of the experiment ($n = 1$), leaving a sample of $n = 170$ (age: $M = 23.25$, $SD = 2.96$, range: 18–35; 98 female participants, 72 male participants). Together with the relevant data collected in study 1 (i.e. the two groups 'human/label human' and 'AI/label human'), the final sample size of study 2 was $n = 334$ (age: $M = 23.22$, $SD = 2.98$, range: 18–35; 194 female participants, 140 male participants). The total sample size of both studies was $n = 492$ (age: $M = 23.38$, $SD = 3.11$, range: 18–35; 266 female participants, 226 male participants).

### Procedure

**Preparation of human-generated responses.** Participants engaged in a text-based online version of the FFP, designed to generate interpersonal

closeness between two unfamiliar interaction partners through escalating mutual self-disclosure in a series of turn-taking questions[36]. Human responses were generated in two in-person laboratory sessions in which individuals of the same gender took part simultaneously (session 1: 10 women; session 2: 10 men; $n = 20$; age: $M = 21.60$; $SD = 2.11$; range: 19–25). Note that in the main online experiment, the age of all human interaction partners was standardised to 25 to match the age of AI-generated partners, as similarity in age can influence perceived closeness. Participants were recruited with flyers and public notices and were pre-screened via an online screening questionnaire to assess exclusion criteria, which were analogous to the criteria of the two main studies. They then responded to three warm-up items on basic character information (i.e. 'What is your name and how old are you?'; 'Where do you live?'; 'What do you study?'), followed by the eight small-talk items and the eight deep-talk items from the FFP, as required for the respective conditions in the main online experiment (all items available in Supplementary Table 1). Participants were instructed to respond within 90 s to each item to enable short, spontaneous responses. No actual time limit was set to ensure complete responses. Participants in the laboratory sessions were informed that they would be matched with another person in the same room, and that each interaction partner would receive the responses of the other after the experiment. We chose this procedure for three reasons. First, we wanted participants to have the experience of communicating with a real person, thereby maximising their motivation to respond in a way they would find accurate in a social interaction. Second, we wanted to parallelize the procedure of this appointment with that of the online experiment. Third, in order to use laboratory-generated answers in the online experiment, it was necessary to avoid references to previous responses of the partner that would not be meaningful in the context of other interactions.

Participants in the in-person laboratory sessions were compensated with €20 plus additional earnings from a Trust-Game (which was no further analysed here due to extreme ceiling effects, for details see section 'Measures of relationship building') of $M = €3.13$ on average ($SD = 0.886$). After the experiment, they received the responses of their interaction partner to the FFP items. From the total sample, the responses of three men and three women were randomly selected as the interaction partner's responses for the online experiment. Although only three sets of responses were needed from each group, we invited 10 participants to each laboratory session to enhance the feeling of actual interactions and to account for potential dropouts due to inadequate responses. In the end, none of the participants provided inadequate responses.

**Preparation of AI-generated responses.** AI responses were generated using the LLM PaLM 2 (interface: Google BARD; Google LLC, CA, USA; date of access: February 19, 2024). The AI was instructed to create six fictional student characters (three men and three women) aged 25 and then answer the eight small-talk and the eight deep-talk items of the FFP from the perspective of these fictional characters. As we found that the plausibility of AI-generated characters decreased as their number increased, we set six as a compromise between plausibility and representability. The following minimal prompt was used to keep AI responses as close as possible to its default style: 'Create six different biographies of typical students (three women and three men) aged 25 and answer the following questions from the perspective of the six students.' Importantly, responses of all AI-generated partners were generated in a single session to ensure character consistency across FFP items. The age of 25 was chosen as it was the approximate average age expected for the study sample, ensuring minimal age difference (potentially affecting relationship building) with participants ranging from 18 to 35 years. Note that responses to warm-up items for both human and AI interaction partners were drawn from the aforementioned AI-generated characters to maintain consistency across conditions in terms of name, age, place of residence, and field of study. This approach ensured that any observed differences between conditions could be attributed to variations in

responses rather than demographic differences. Biographies of the six AI-generated characters and responses of both AI-generated characters and the six human interaction partners are shown in Supplementary Table 1.

**Recruitment.** To recruit participants, we contacted student councils across German universities, requesting them to forward the study flyer to their students. We also used flyers, public notices and print and online social media to recruit participants in Freiburg, Germany. Via a URL or QR code, interested persons were forwarded to a screening questionnaire, where exclusion criteria were assessed. Suitable participants immediately continued with the online experiment after finishing the screening.

**Online experiment.** This study was approved by the Ethics Committee of the University of Freiburg (ETK-Freiburg application code: 23-1479-S2, February 15, 2024) and conducted in accordance with the principles of the Declaration of Helsinki. No data on race or ethnicity was collected. Data for both study 1 and study 2 were collected simultaneously to prevent history effects (data collection period: March 3 to June 12, 2024). Participants first read and signed an informed consent form, which provided details about the study's procedure and specified whether they would be interacting with another human (studies 1 and 2) or an AI (study 2). They then entered a virtual waiting room for 50 s, during which they were informed that another participant of the same gender was being located for their upcoming online interaction and that this process might take a few minutes. As this study focussed on friendly, non-romantic relationships, participants were always assigned to an interaction partner (real or fictional) of the same gender. Participation in the online study was limited to the hours between 4:00 and 9:00 pm to enhance the plausibility of a real-time interaction. Prior to starting the FFP, participants were instructed to respond to each item within 3 min to avoid lengthy waiting times for their interaction partner. However, no actual time limit was enforced. During the FFP, participants first responded to three warm-up items and, after each response, read the corresponding response to the same item from their interaction partner. To ensure consistency in basic character information across conditions, participants in both the human and AI interaction groups received AI-generated responses to the warm-up items, while no personal information from human partners was used. Next, participants completed a brief baseline questionnaire measuring perceived interpersonal closeness to their partner. They then proceeded with the main FFP phase, consisting of either eight small-talk or eight deep-talk items. Here, they were required to read the partner's response to each item for a minimum of 25 s before they could continue. To enhance the feeling of a live interaction, we introduced random waiting periods of 3 to 10 s after participants submitted their responses. This setup was designed to create the impression that sometimes participants submitted their responses first (resulting in a waiting period), while at other times, their partner responded first (resulting in no waiting period). To reduce potential pressure from sending responses slower than the partner, waiting periods were included slightly more often than not (i.e. in 5 out of 8 cases). After responding to the last item, participants completed the brief questionnaire again to assess perceived interpersonal closeness to their interaction partner after completion of the FFP. Next, participants engaged in an interactive Trust Game where they could earn additional monetary gains based on their own decisions and those of their interaction partner[46]. Finally, participants completed questionnaires measuring individual trait characteristics evaluated elsewhere. Two attention control questions were included in the questionnaire battery to identify inattentive participants, explicitly requesting them to select a specific response option. In total, the experiment had a duration of ~40 min. Participants of each study received a compensation of €15 plus the additional average gain from the Trust Game of $M = €3.54$ ($SD = 0.969$). As this research involved a degree of deception about the true source of the responses and the real-time nature of the social interaction, participants were debriefed via email about the true source of the responses they had read and about the fact that the responses had been generated before the actual experiment took place. Debriefing took place 2 weeks after the end of data collection.

### Measures of relationship building

To obtain an intuitive measure of perceived interpersonal closeness to the interaction partner (or simply 'interpersonal closeness'), we applied an adapted version of the widely used Inclusion of Other in the Self Scale (IOS[47–49]), a one-item pictorial scale including nine images depicting two progressively overlapping circles to represent the 'self' and the 'other'. Here, we labelled the 'other' as the 'interaction partner' to specifically measure closeness to the interaction partner. More overlapping circles indicate a higher 'inclusion of the other in the self' and thus a higher perceived interpersonal closeness. The IOS shows good psychometric properties, including 2-week retest-reliability ($r = 0.86$) and convergent, discriminant and predictive validity[47,50]. Note that we refrained from using difference scores (post minus pre) in our main analyses, as pre-measures are likely already influenced by the information from the warm-up items of the FFP (i.e. name, age, residence, and field of study), which can bias social perception[51,52]. Subtracting these initial impressions could thus reduce the variance of interest in the post-measures of closeness. This issue is further amplified in the two 'label AI' conditions, where pre-measures are additionally shaped by participants' expectations of interacting with AI, meaning that differencing would remove precisely those initial attitudes that are central to our research question. Consistent with our pre-registration, we therefore relied on post measures only.

We pre-registered the use of the Interpersonal Closeness Questionnaire[53] as an additional measure of interpersonal closeness. However, we chose to focus on the IOS for our analyses because (a) the two measures showed high redundancy ($r_{(490)} = 0.723$, $p < 0.001$, 95% CI [0.678, 0.762]), (b) the IOS is more widely used than the ICQ, and (c) the IOS was more sensitive to the FFP across both studies and all conditions (as indicated by increases from pre- to post-measures analysed in hierarchical linear models for repeated measures [for details, see section 'Statistical analyses']; IOS: $t_{(493)} = 19.40$, $p < 0.001$, $R^2_m = 0.106$; ICQ: $t_{(493)} = 14.50$, $p < 0.001$, $R^2_m = 0.072$). Note that the key results of both studies are similar when using the ICQ. We also planned to include trust towards the interaction partner, assessed in an interactive decision-game with real monetary consequences (i.e. the Trust Game[46]), as a measure of relationship building. However, this measure exhibited limited variability due to extreme ceiling effects, with 47.4% of participants choosing the maximum transfer. As a result, we decided not to include the analysis of trust in the study.

### Linguistic analysis

To explore potential reasons for identified differences in relationship building between treatment groups in both studies, we conducted automated linguistic analyses of participants' as well as their human or AI interaction partners' responses to FFP items using the Linguistic Inquiry and Word Count system (LIWC-22[39,40]). The LIWC is a software that counts words in text files that match specific categories and quantifies them relative to the total word count. In the current study, we focused on analysing self-disclosure[41], which is assumed to be the main mechanism underlying the establishment of interpersonal closeness in the FFP[36]. In the LIWC, the category self-disclosure consists of the three subscales 'self-related personal pronouns' (e.g. I, me, my), 'emotion' (e.g. happy, cry, abandon), and 'social processes' (e.g. friend, talk, family), together forming the self-disclosure variable. Consistent with Callaghan and colleagues[41], who developed the self-disclosure measure within the LIWC, we also analysed the total response length as an indicator of the motivation to engage in the social interaction.

### Statistical analyses

Research questions and analyses were pre-registered at the OSF repository (https://osf.io/chdx7). The data and code used to generate the results of this study are freely available there as well (https://osf.io/qs6yf/). Closeness measures after the interaction (i.e. post measures) were used as the main

dependent variable across both studies. As basic analyses in both studies, we computed hierarchical linear models (HLM) for repeated measures to investigate whether any increase in interpersonal closeness would take place over the course of the FFP in specific experimental conditions (e.g. in masked deep-talk interactions with AI [study 1, pre-registered hypothesis 1] and unmasked deep-talk interactions with AI [study 2, pre-registered hypothesis 5]). These analyses included the repeated measures factor 'time' (two levels 'pre' and 'post' per participant), the dependent variable 'interpersonal closeness', the covariates age and gender, and a random intercept across participants (as pre- and post-measures of closeness were nested within participants). To measure effect size in these models, we computed marginal R-squared values following the recommended procedure by Nakagawa & Schielzeth[54]. To test whether increases in closeness differed meaningfully between human and AI interactions, we conducted equivalence testing (TOST) with bounds of ±0.35 pooled standard deviations, representing small-to medium effect sizes. This analysis was complemented by a Bayesian two-sample $t$-test to quantify evidence for or against a meaningful difference.

Participants were nested within groups, each interacting with one out of 24 partners (2 source identities [human vs. AI] × 2 emotional intensities [small-talk vs. deep-talk] × 2 genders [female vs. male] × 3 human or AI-generated partners per condition). To assess data dependency in interpersonal closeness ratings, we used an ANOVA to compare a standard model of closeness with a random intercept model allowing variation across interaction partners[55]. As the model with random variation across intercepts did not surpass the standard model ($p = 0.356$), HLM was not required for our main analyses. Thus, ANCOVAs were conducted to test main and interaction effects of the independent variables 'source identity' (human vs. AI; studies 1 and 2), 'emotional intensity' (small-talk vs. deep-talk; study 1), and 'source label' (label human vs. label AI, study 2) on the dependent variable 'interpersonal closeness' while controlling for the effects of age and gender (pre-registered hypotheses 2, 3, 4 [study 1], and 6, 7, and 8 [study 2]). Analogous analyses were performed within specific treatment groups as post-hoc tests. Whenever relevant, we formally tested the normality of residuals and equality of variances. The assumption of equal variances was met in all analyses, as indicated by Levene's test (all $p > 0.05$). Although interpersonal closeness was not normally distributed across studies (Shapiro–Wilk test, $p < 0.001$), and residuals in some analyses deviated from normality, both HLM and ANOVA were applied, as they are generally robust to violations of normality, especially with large sample sizes[56,57].

We then investigated whether treatment group differences in closeness could be explained by differences in self-disclosure and/or response length. Specifically, we conducted (a) ANCOVA to determine whether treatment group differences in closeness would parallel treatment group differences in the participant's and/or their interaction partner's self-disclosure and/or response length, and (b) partial correlation analyses to examine whether the participant's and/or their interaction partner's self-disclosure and/or response length were associated with the participant's perceived interpersonal closeness. Age and gender were controlled for in both analyses. $P$-values smaller than 0.05 (two-tailed) were considered statistically significant across all analyses.

## Results
### Do people build human-like relationships with AI?
As a basic analysis of study 1, we first tested whether interpersonal closeness increased at all in masked interactions with an AI (i.e. the interaction partner was labelled as a human). Repeated measures HLM analyses revealed statistically significant increases in closeness from pre- to post-interaction measures in human-labelled AI interactions across both deep-talk and small-talk conditions ($t_{(165)} = 11.92$, $b = 1.16$, 95% CI [0.970, 1.36], $p < 0.001$, $R^2_m = 0.121$; see Supplementary Fig. 1A) and separately within each condition (small-talk: $t_{(81)} = 7.26$, $b = 0.988$, 95% CI [0.717, 1.26], $p < 0.001$, $R^2_m = 0.112$; deep-talk: $t_{(83)} = 9.67$, $b = 1.33$, 95% CI [1.06, 1.61], $p < 0.001$, $R^2_m = 0.153$). As predicted, deep-talk interactions with AI significantly increased interpersonal closeness compared to a baseline measure,

confirming hypothesis 1. For comparison, we also analysed the increase in closeness in interactions with humans. Again, we found statistically significant increases in closeness across both deep-talk and small-talk conditions ($t_{(155)} = 11.25$, $b = 1.10$, 95% CI [0.909, 1.30], $p < 0.001$, $R^2_m = 0.114$; see Supplementary Fig. 1B) and within each condition (small-talk: $t_{(75)} = 7.47$, $b = 1.13$, 95% CI [0.830, 1.43], $p < 0.001$, $R^2_m = 0.100$; deep-talk: $t_{(79)} = 8.48$, $b = 1.07$, 95% CI [0.823, 1.33], $p < 0.001$, $R^2_m = 0.138$). An equivalence test followed by a Bayesian $t$-test revealed that differences in closeness increases between human and AI interactions across levels of emotional intensity were practically negligible when testing against the presence of a small-to-medium effect ($t_{(139.58)} = -2.71$, $p = 0.004$, Hedges' $g = 0.048$, 95% CI [−0.170, 0.266]). The Bayes-Factor, using the default medium-sized JZS prior ($r = 0.707$; $BF_{01} = 7.43$; posterior mean effect size $\delta = 0.046$, based on 20,000 iterations; 95% credible interval [−0.168, 0.258]), indicated that the data were 7.43 times more likely to occur under the null hypothesis than under the alternative hypothesis. To assess robustness, we repeated the analysis with a wider prior ($r = 1.00$), resulting in a similar conclusion ($BF_{01} = 10.37$).

As the main analysis of study 1, we compared interpersonal closeness as measured after FFP interactions by running an ANCOVA with the independent variables 'source identity' (levels: source human and source AI), 'emotional intensity' (levels: small-talk and deep-talk), their interaction, and the covariates age and self-reported gender. While there were no statistically significant main effects of 'source identity' ($F_{(1, 316)} = 0.097$, $p = 0.756$, $\eta^2_p < 0.001$, 95% CI [0.00, 0.014]; $M_{source\ human} = 4.08$, $SD = 1.71$; $M_{source\ AI} = 4.15$, $SD = 1.94$) or 'emotional intensity' ($F_{(1, 316)} = 0.420$, $p = 0.518$, $\eta^2_p = 0.001$, 95% CI [0.00, 0.021]; $M_{small-talk} = 4.06$, $SD = 1.90$; $M_{deep-talk} = 4.17$, $SD = 1.77$), we found a statistically significant interaction effect between the two variables ($F_{(1, 316)} = 8.17$, $p = 0.005$, $\eta^2_p = 0.025$, 95% CI [0.002, 0.068]). Post-hoc tests revealed higher closeness after interacting with an AI compared to a human within the deep-talk condition ($F_{(1, 160)} = 4.89$, $p = 0.028$, $\eta^2_p = 0.030$, 95% CI [0.00, 0.096]; $M_{source\ human} = 3.85$, $SD = 1.69$; $M_{source\ AI} = 4.48$, $SD = 1.80$), but no statistically significant difference in closeness within the small-talk condition ($F_{(1, 162)} = 2.89$, $p = 0.091$, $\eta^2_p = 0.018$, 95% CI [0.00, 0.078]; $M_{source\ human} = 4.33$, $SD = 1.71$; $M_{source\ AI} = 3.82$, $SD = 2.04$; see Fig. 1A). Surprisingly, our results provide neither evidence that interactions with humans yield stronger feelings of closeness than interactions with AI, nor that deep-talk interactions yield stronger feelings of closeness than small-talk interactions. Consequently, hypotheses 2 and 3 are rejected. Furthermore, and contrary to hypothesis 4, AI interactions yielded stronger feelings of closeness than human interactions, but only in the deep-talk condition, not in the small-talk condition.

To further explore why people feel closer to the AI than to humans following deep-talk interactions, we tested whether AI-generated responses differed from human-generated responses regarding self-disclosure as measured by the Linguistic Inquiry and Word Count system (LIWC-22[39,40]). An ANOVA (using a dataset including all human- and AI-generated responses within the deep-talk condition; $n = 12$) revealed that AI-generated responses showed considerably higher levels of self-disclosure than human-generated responses ($F_{(1, 10)} = 18.57$, $p = 0.002$, $\eta^2_p = 0.650$, 95% CI [0.167, 0.801]; $M_{AI} = 38.03$, $SD = 2.62$; $M_{human} = 32.31$, $SD = 1.92$; Fig. 1B; no statistically significant difference regarding response length; $F_{(1, 10)} = 0.802$, $p = 0.392$, $\eta^2_p = 0.074$, 95% CI [0.000, 0.407]; $M_{AI} = 313.83$, $SD = 18.89$; $M_{human} = 295.50$, $SD = 46.45$). Next, we tested whether self-disclosure shown in partner responses was related to participants' perceived interpersonal closeness using partial correlation analysis controlling for age and gender (using a dataset of deep-talk interactions in study 1; $n = 164$). We found that participants felt closer to their interaction partner when their partners' responses displayed higher levels of self-disclosure ($r_{p(160)} = 0.242$, 95% CI [0.091, 0.382], $p = 0.002$, $R^2 = 0.058$; Fig. 1C).

We then tested whether the participants' own responses differed between interactions with an AI and interactions with a human. An ANCOVA demonstrated that participants showed higher levels of self-disclosure in interactions with an AI ($F_{(1, 160)} = 3.92$, $p = 0.049$, $\eta^2_p = 0.024$,

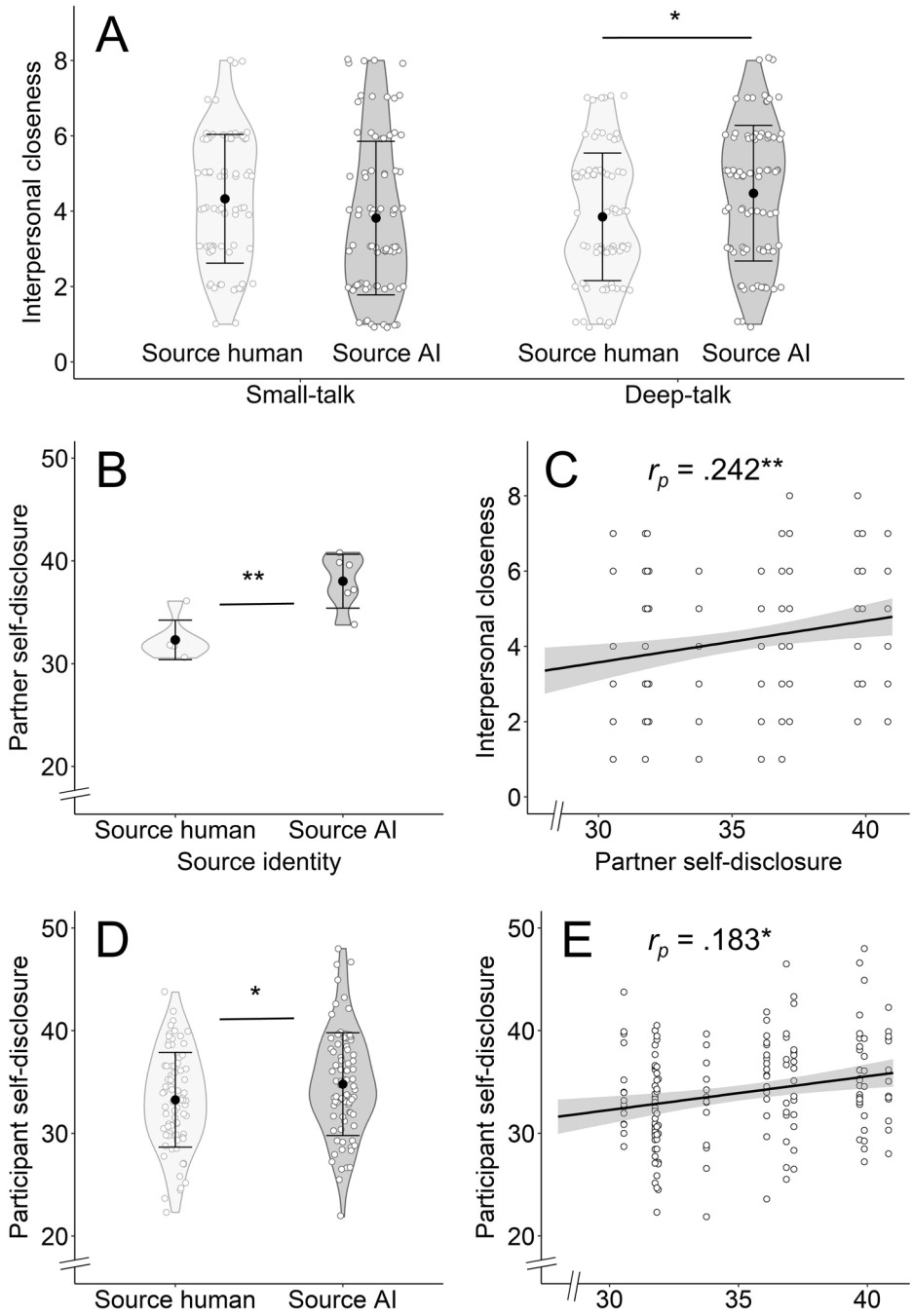

**Fig. 1 | AI-generated content leads to higher levels of interpersonal closeness than human-generated content in deep-talk interactions.** $^*p < 0.05$, $^{**}p < 0.010$, $^{***}p < 0.001$. **A** Violin plots of perceived interpersonal closeness in the four different conditions of study 1 ($n = 322$). There was significantly higher interpersonal closeness following interactions with AI compared to interactions with humans within deep-talk interactions. **B** Violin plots of self-disclosure (as measured via the LIWC-22) displayed by the six human and the six AI interaction partners within the deep-talk condition ($n = 12$). AI interaction partners showed significantly more self-disclosure. **C** Scatterplot showing a statistically significant positive association between the partner's self-disclosure and interpersonal closeness ($n = 164$). Participants felt closer to their partner when the partner showed more self-disclosure.

**D** Violin plots of participants' own self-disclosure when interacting with a human or with an AI ($n = 164$). Participants showed significantly more self-disclosure when interacting with an AI. **E** Scatterplot showing a statistically significant positive association between the partner's self-disclosure and participants' own self-disclosure ($n = 164$). Participants showed more self-disclosure when their partner showed more self-disclosure. This suggests that participants tend to disclose more personal information to AI interaction partners in response to the AI's higher level of self-disclosure, indicating a reciprocal effect. All violin plots include individual data points, means and standard deviations. All scatterplots include individual data points, regression slopes, and 95% confidence intervals.

95% CI [0.00, 0.087]; $M_{AI} = 34.80$, $SD = 5.00$; $M_{human} = 33.26$, $SD = 4.61$; Fig. 1D; no statistically significant difference regarding response length; $F_{(1, 160)} = 0.007$, $p = 0.932$, $\eta^2_p < 0.001$, 95% CI [0.00, 0.012]; $M_{AI} = 197.62$, $SD = 62.58$; $M_{human} = 199.15$, $SD = 64.58$). However, participants own

self-disclosure was not significantly associated with their perceived closeness in deep-talk interactions ($r_{p(160)} = 0.117$, 95% CI [−0.038, 0.266], $p = 0.139$). Lastly, we analysed whether the degree of self-disclosure shown in the responses of the interaction partners related to participants' own

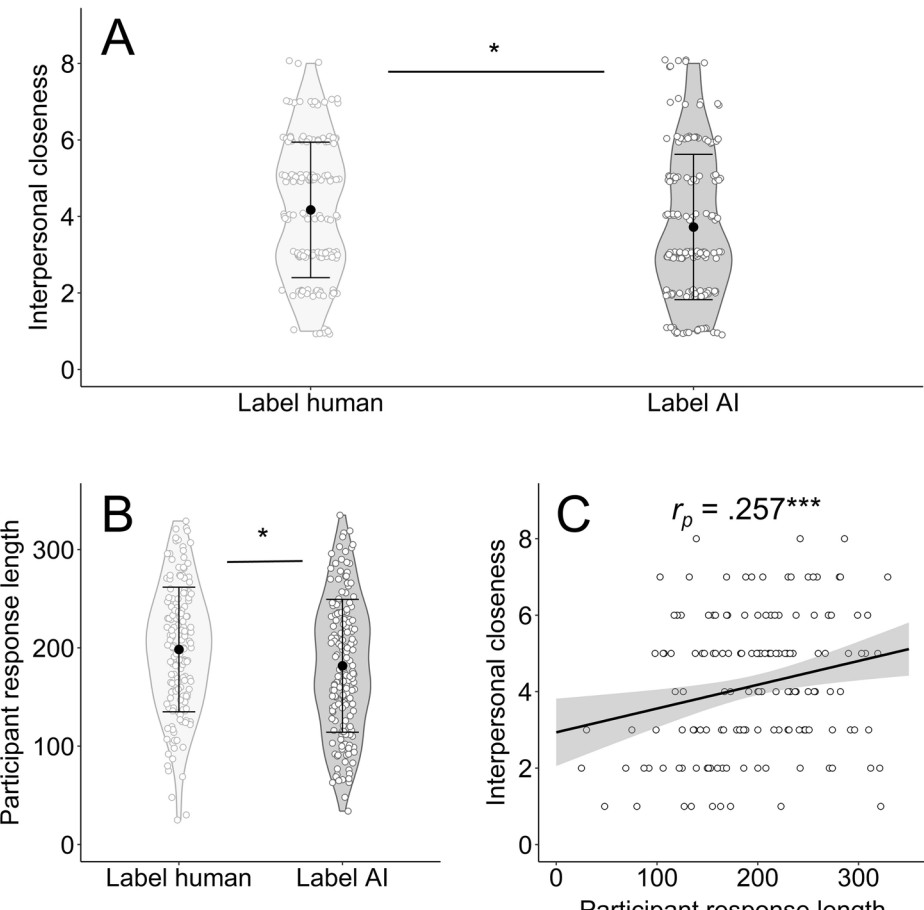

**Fig. 2 | The anti AI-bias.** $^{*}p < 0.05$, $^{**}p < 0.010$, $^{***}p < 0.001$. **A** Violin plots of perceived interpersonal closeness following AI- and human-labelled interactions ($n = 334$). Participants who believed they would interact with an AI reported significantly lower interpersonal closeness than participants who believed they would interact with a human. **B** Violin plots of the participants' response length in AI- and human-labelled interactions ($n = 334$). Participants who believed they would interact with an AI wrote significantly shorter responses than participants who

believed they would interact with a human. All violin plots include individual data points, means and standard deviations. **C** Scatterplot with individual data points, regression slope, and 95% confidence interval, illustrating a statistically significant positive association between the participants' response length and perceived interpersonal closeness ($n = 334$). Participants who wrote longer responses felt closer to their interaction partner.

self-disclosure. Indeed, participants' self-disclosure was positively associated with their interaction partner's self-disclosure ($r_{p(160)} = 0.183$, 95% CI [0.030, 0.328], $p = 0.020$, $R^2 = 0.033$; Fig. 1E).

To summarise the findings of study 1, the AI-generated content outperformed human-generated content in establishing feelings of closeness during emotionally engaging deep-talk interactions. This effect appears to be driven by higher levels of self-disclosure by AI partners compared to human partners, which, in turn, enhanced participants' perceived interpersonal closeness. Moreover, participants disclosed more information themselves in interactions with AI, and self-disclosure levels of both parties were associated with each other. These findings suggest that the AI's increased self-disclosure motivates participants to disclose more personal information themselves, ultimately leading to more intimate interactions and a stronger sense of closeness.

### Does the belief of interacting with an AI hinder relationship building (study 2)?

Building on the basic analysis from study 1, which demonstrated that relationship building occurs in masked interactions with an AI (i.e. when the AI was labelled as a human), we first examined in study 2 whether the belief of interacting with an AI prevents relationship building. Using repeated measures HLM analyses as described before, we analysed whether interpersonal closeness increased in interactions in which participants were informed they would interact with an AI, both when they actually interacted

with the AI ('source AI/label AI') and when they actually interacted with a human ('source human/label AI'). Repeated measures HLM analyses revealed statistically significant effects of 'time' (pre vs. post interaction) on closeness both across levels of 'source identity' ($t_{(169)} = 10.41$, $b = 1.03$, 95% CI [0.834, 1.22], $p < 0.001$, $R^2_m = 0.120$; see Supplementary Fig. 1C) and separately within each condition (interactions with humans: $t_{(83)} = 6.93$, $b = 1.03$, 95% CI [0.738, 1.33], $p < 0.001$, $R^2_m = 0.187$; interactions with AI: $t_{(82)} = 7.91$, $b = 1.02$, 95% CI [0.767, 1.28], $p < 0.001$, $R^2_m = 0.080$). These findings demonstrate that relationship building occurred even when participants were informed they were interacting with an AI, supporting hypothesis 5. For comparison, we also analysed the increase in closeness in interactions in which participants were informed they would interact with a human (across levels of 'source identity'). The analysis revealed a statistically significant increase in closeness across sources when participants were informed they would interact with a human ($t_{(163)} = 12.82$, $b = 1.21$, 95% CI [1.02, 1.39], $p < 0.001$, $R^2_m = 0.123$; see Supplementary Fig. 1D).

As the main analysis of study 2, we compared closeness after AI-labelled and human-labelled interactions by running an ANCOVA with the independent variables 'source identity' (pre-registered hypothesis 6), 'source label' (pre-registered hypothesis 7), their interaction (pre-registered hypothesis 8), and the covariates age and gender. The analysis revealed a statistically significant main effect of 'source label', with lower interpersonal closeness following AI-labelled interactions ($F_{(1, 328)} = 4.90$, $p = 0.028$, $\eta^2_p = 0.015$, 95% CI [0.00, 0.050]; $M_{label\ human} = 4.17$, $SD = 1.77$,

$M_{label\ AI} = 3.72$, $SD = 1.90$; Fig. 2A), but no statistically significant main effect of 'source identity' ($F_{(1,\ 328)} = 3.05$, $p = 0.082$, $\eta^2_p = 0.009$, 95% CI [0.00, 0.041]; $M_{source\ human} = 3.77$, $SD = 1.79$, $M_{source\ AI} = 4.11$, $SD = 1.90$) and no statistically significant interaction ($F_{(1,\ 328)} = 2.03$, $p = 0.155$, $\eta^2_p = 0.006$, 95% CI [0.00, 0.033]). Again, these results provide no evidence that participants establish stronger feelings of closeness with human interaction partners than with AI interaction partners. Thus, hypothesis 6 is rejected. As expected, labelling the interaction partner as an AI led to lower interpersonal closeness ratings after the interaction compared to when the partner was labelled as human, demonstrating an anti-AI bias and confirming hypothesis 7. Hypothesis 8 was rejected, as we found no evidence for differences in interpersonal closeness (human > AI) following human interactions compared to AI interactions.

As participants were presented with responses from the same human or AI interaction partners in both human-labelled and AI-labelled conditions, the difference in interpersonal closeness cannot be attributed to variations in the partners' responses. We therefore tested whether people themselves communicated differently with AI-labelled partners than human-labelled partners in an exploratory fashion. Indeed, while there were no statistically significant group differences regarding self-disclosure ($F_{(1,\ 330)} = 0.050$, $p = 0.823$, $\eta^2_p < 0.001$, 95% CI [0.000, 0.012]; $M_{AI} = 34.14$, $SD = 4.39$; $M_{human} = 34.05$, $SD = 4.86$), people wrote significantly longer responses when assuming they would interact with a human ($F_{(1,\ 330)} = 5.32$, $p = 0.022$, $\eta^2_p = 0.016$, 95% CI [<0.001, 0.052]; $M_{AI} = 181.78$, $SD = 67.55$; $M_{human} = 198.37$, $SD = 63.37$; Fig. 2B). In turn, longer responses were also related to higher levels of perceived closeness ($r_{p(330)} = 0.257$, 95% CI [0.154, 0.355], $p < 0.001$, $R^2 = 0.066$; Fig. 2C). AI-labelled interactions thus resulted in both shorter responses of participants and lower levels of perceived closeness.

We found evidence that individuals form social bonds with AI even when being aware of interacting with an artificial agent, yet also observed an anti-AI bias leading to lower feelings of closeness after the interaction compared to human-labelled interactions. To explore why some people form social bonds to AI while others do not, we examined whether AI scepticism is more pronounced in individuals who value natural human communication. Indeed, we found a statistically significant interaction effect between 'source label' and *universalism*, a personal value centred on concern for the welfare of people and nature[58], in predicting interpersonal closeness ($F_{(1,\ 328)} = 4.11$, $p = 0.043$, $\eta^2_p = 0.012$, 95% CI [0.00, 0.046]). This effect was driven by a positive association between universalism and closeness in human-labelled interactions ($r_{p(162)} = 0.240$, 95% CI [0.136, 0.339], $p = 0.002$, $R^2 = 0.058$), which was not present in AI-labelled interactions ($r_{p(168)} = 0.017$, 95% CI [−0.091, 0.125], $p = 0.825$, $R^2 < .001$). These findings indicate that personal values may modulate relationship building with AI.

To summarise the findings of study 2, relationship building occurred even when participants believed that they interacted with AI. However, participants felt less close to AI-labelled partners compared to human-labelled partners. This effect seems to stem from lower motivation to engage with AI partners, as evidenced by shorter responses.

## Discussion

Can we 'befriend' an AI? The present study examined the effects of source identity (AI-generated responses vs. human-generated responses), source label (interaction partner labelled as an AI vs. interaction partner labelled as a human), and emotional intensity (small-talk vs. deep-talk) on relationship building. We found that humans indeed build relationships with fictional characters created by AI. Strikingly, AI-generated content outperformed human-generated content in establishing feelings of closeness during emotionally engaging deep-talk interactions (including topics such as the most treasured memory of your life or what you value most in a friendship). Follow-up analyses suggest that AI-responses featured higher levels of self-disclosure in emotional interactions, which in turn elicited higher levels of self-disclosure and perceived interpersonal closeness in participants. Being explicitly informed that one would interact with an AI led to an anti-AI bias, which reduced, but did not prevent, relationship building. This effect might

be due to lower motivation of participants to engage with AI, as evidenced by both shorter responses and lower levels of reported closeness.

One of the key innovations of our study is the direct comparison between human-to-AI and human-to-human relationship building. Recent studies have demonstrated that people do form some sort of relationship with 'AI social companions'[59], and highlight AI's socio-emotional capabilities, such as emotional awareness[27], empathetic responses[60], and the ability to offer relationship advice[11]. Expanding upon this research, we provide evidence that, when assuming they are interacting with a human, people form relationships with AI to a similar extent as with fellow humans, as shown by a similar increase in interpersonal closeness over the course of the interaction. Moreover, we found that people felt even closer to AI than to fellow humans after emotionally engaging interactions. At first glance, this seems counterintuitive, as AI lacks the emotion-related bodily sensations that underpin human emotional experiences[61]. On second thought, however, this 'deficit' could also create an advantage for AI. For humans, disclosing personal information on emotionally charged topics is risky and requires trust, as the recipient might fail to show the desired empathic reaction or even misuse the information to one's disadvantage. As a result, humans often avoid discussing emotional topics to protect themselves[62,63]. In contrast, as AI cannot experience emotions, there are no such restrictions when opening up about emotionally charged topics. Indeed, follow-up linguistic analyses suggest that AI-generated responses showed higher levels of self-disclosure, which, in turn, also encouraged more self-disclosure by participants[41]. Importantly, both the partners' and participants' own self-disclosure were associated with higher levels of perceived interpersonal closeness. Thus, our findings highlight not only AI's ability to excel in emotionally charged communication, but also its potential to help humans feel more comfortable opening up compared to interactions with another human.

Does the higher level of self-disclosure shown by AI imply that AI is generally superior to humans in emotional conversations? Probably not. First, emotional conversations between humans serve many purposes, and building a relationship is just one of them. As neither human nor AI responses in our study were generated with a specific goal in mind, humans may still outperform AI when actively trying to build a relationship. Second, high self-disclosure at certain levels and in certain situations may be perceived as unnatural (e.g. when interacting with complete strangers), unprofessional (e.g. in the workplace), or even risky (e.g. when interacting with an untrustworthy person[62,63]). AI models may be less flexible and precise than humans in appropriately adjusting self-disclosure across different situations. Third, AI-generated content only resulted in greater feelings of closeness when disguised as human-generated, which is not the case in everyday applications. Consistent with recent findings showing that labelling content as AI-generated reduces its positive perception[7], participants reported less interpersonal closeness when they were informed they would interact with an AI. This is not surprising, as humans generally prefer human over AI partners, particularly in areas often considered uniquely 'human' such as emotional interactions[25,26,64,65]. Additional analyses showed that participants provided shorter responses under these circumstances, indicating less motivation to engage in personal interactions with an AI. These findings align with the Social Need Fulfilment Model for Human-AI relationships[66], which suggests that human-AI interactions typically satisfy only concrete social needs (e.g. pleasure) rather than deeper, symbolic needs (e.g. feeling genuine care). Humans possess a fundamental, evolutionary need for social connection that involves shared emotions, mutual understanding, and the comfort of knowing someone else truly comprehends our feelings[67,68]. The knowledge that another person feels what we feel, understands our intentions, and responds with empathy and authenticity is essential to humanity, something that AI, at least at the current state, cannot truly replicate. So why do individuals form relationships with AI at all, even when being aware of its artificial nature (also see ref. 59)? One explanation is the phenomenon of anthropomorphism, the tendency to attribute human traits, emotions, or intentions to non-human entities[16,17]. Humans are inherently social, so when presented with AI-generated cues that resemble

human-generated cues, the brain may intuitively respond to these artificial cues much as it would to genuine social cues. At the same time, our results highlight that at least some individuals remain reluctant to engage with AI, leading to the question of how these people differ from those who are more receptive. In an exploratory analysis, we found that universalism moderated the difference in interpersonal closeness following AI- versus human-labelled interactions. Specifically, individuals high in universalism felt closer to humans, but not to AI. This indicates that traits linked to natural social interaction, and, vice versa, potentially to negative attitudes towards artificial interaction, may reduce the likelihood of forming bonds with AI. However, this result requires further validation in future research.

As has often been the case with technological advances throughout history[69,70], the rise of AI brings both benefits and risks to society. Our findings underscore these two sides of the coin in social applications, highlighting AI's potential for relationship building in overburdened social fields, while also emphasising the risks of its misuse, especially when disguised as human. Healthcare is struggling to meet the growing demand for services due to factors such as aging populations, reduced funding, and the growing psychological toll on healthcare workers[71–73]. Conversational AI may help alleviate this burden. As demonstrated by our finding that AI excels particularly in emotional conversation, it could be effective in psychotherapy and medical settings where relationship building and adequate interaction on emotional topics is key (for reviews, see refs. 74,75). These settings include health- or psychoeducation, providing care for individuals with limited access to therapy, offering social contact to alleviate loneliness in the elderly, bridging waiting times until the start or between psychotherapy sessions, and facilitating communication with patients[75–78]. Importantly, AI should assist, not replace, therapists, as exclusive reliance on AI for addressing health issues may lead to over-reliance, addiction, or withdrawals from human relationships[79–82], as well as other unforeseeable harmful effects (e.g. not adequately reacting to the expression of suicidal intentions[83]). Therefore, a human introduction and ongoing monitoring are imperative for safe and effective use. Our results also demonstrate negative attitudes towards social interaction with AI, even in a relatively young sample aged 18 to 35 years. To fully unlock AI's potential in healthcare applications, efforts are needed to increase AI acceptance in these fields, e.g. by providing a clear and transparent explanation of the reasons for using AI by a human before an AI intervention begins, or by implementing follow-up human-to-human sessions to discuss and integrate previous human-to-AI interactions[84–88].

Beyond AI's potential benefits in healthcare, our findings also highlight the risks AI poses for society, especially when AI-generated information is presented as human-created. AI-generated content is already flooding social media[89]. With the quality of AI content improving to the point where it even outperforms humans in certain social contexts (such as relationship building in emotional interactions), the risk of people falling for deceptive traps continues to grow. Specifically, AI's ability to build social-emotional relationships can be misused to establish deceptive emotional connections, steal personal data, and enable exploitation by unethical actors, including both individuals and corporations[90–92]. Obviously, people are far more likely to buy products, disclose personal information, or send money when a request appears to come from a friend rather than from an easily recognisable scam email. Effectively tackling AI misuse requires a combination of transparent regulations, ethical guidelines, robust detection mechanisms, and public awareness to ensure responsible development and use[85,93].

## Limitations

We acknowledge several limitations of this study. Building on our findings, future studies could explore relationship building with AI in real-time interactions by combining conversational AI with avatars and voice generation[94,95]. However, while this could enhance efficiency and acceptance by making interactions become more life-like, it may also backfire if AI becomes too human-like as suggested by the Uncanny Valley hypothesis[96]. Furthermore, while the FFP used here provides a well-stablished and

effective framework for establishing relationships in a semi-standardised manner[36], less structured interactions may offer greater ecological validity in future research. AI responses to the FFP items were generated using the LLM PaLM 2 in February 2024. As a result, the findings may not generalize to other AI systems. However, this also suggests that we may have underestimated the communicative capabilities of more advanced models available at the time of submission (May 2025), such as GPT-4 (OpenAI, CA, USA) or Gemini 2.5 (Google DeepMind, CA, USA). The finding that AI outperformed humans in fostering emotional connections, even when using now-outdated software, speaks volumes about the potential of future AI systems in this domain. Additionally, the AI was prompted to respond from the perspective of six students. Although we used only a one-sentence minimal prompt to keep responses as close as possible to standard AI output, this approach may still affect how our findings relate to typical AI interactions, which do not involve such prompting. Importantly, however, the prompt did not include instructions regarding the tone of the interaction (e.g. the degree of self-disclosure, empathy, or emotionality), demonstrating that AI-generated responses showed self-disclosure and fostered relationship building even without specific prompting to do so. As noted by a reviewer, an alternative and less minimal prompt could be to instruct the AI to respond from the perspective of the specific human profiles used in this study, which could provide broader control for the characters presented. Relatedly, including more than six human and AI interaction partners per condition could be beneficial in future studies to better represent typical human and AI responding. We also note that this study features a WEIRD sample (Western, educated, industrialised, rich, and democratic[97]), limiting the generalisability of our findings. However, given that AI technologies are predominantly developed and adopted in WEIRD contexts, these settings provide a meaningful foundation for studying human-AI interactions, even if broader generalisability remains an open question. Finally, longitudinal studies are needed to examine whether human-AI relationships can be sustained over time and whether they can reach or even surpass the well-documented long-term mental and physical benefits of human social bonding[68,98–103].

## Conclusion

In conclusion, we present three key findings: First, people form relationships with AI to a similar extent as with other humans when the partner is labelled as human. Second, even minimally prompted AI can outperform humans in establishing feelings of closeness in emotional conversations, partly due to higher levels of self-disclosure. Third, people show an anti-AI bias, as evidenced by weaker relationship building when the partner is explicitly labelled as AI. Together, these findings highlight the dual role of conversational AI as both a powerful tool and a potential risk for society. On one hand, AI shows great promise in alleviating strain in overburdened social fields such as psychotherapy, medical care, and elder care. To foster acceptance in these areas, we recommend transparent human-led introduction, continuous monitoring, and systematic evaluation of human-AI interactions. On the other hand, our results underscore the risk of AI being misused for manipulation by fostering deceptive emotional connections. Clear ethical guidelines and safeguards are therefore crucial to ensure that conversational AI is leveraged responsibly and for societal benefit.

## Data availability

The data used to generate the results of this study are freely available in the OSF repository (https://osf.io/qs6yf/).

## Code availability

The code used to generate the results of this study are freely available in the OSF repository (https://osf.io/qs6yf/).

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

## Acknowledgements
Funded by the European Union. Views and opinions expressed are however those of the author(s) only and do not necessarily reflect those of the European Union or the European Research Council Executive Agency. Neither the European Union nor the granting authority can be held responsible for them. This work is supported by ERC grant 'From face-to-face to face-to-screen: Social animals interacting in a digital world', SODI, 101076414, https://doi.org/10.3030/101076414, awarded to Bastian Schiller. The funders had no role in study design, data collection and analysis, decision to publish or preparation of the manuscript.

## Author contributions
Conceptualization: T.K., M.W., B.S., M.H.; Methodology: T.K., M.W., J.B., B.S.; Software: T.K., M.W., J.B.; Investigation: T.K., M.W., J.B.; Data Curation: T.K., M.W., J.B.; Formal Analysis: T.K., M.W., B.S.; Resources: M.H., B.S.; Writing – Original Draft: T.K., B.S.; Writing – Review and Editing: M.W., J.B., M.H.; Visualization: T.K.; Supervision: M.H., B.S.; Project Administration: T.K., M.H., B.S.; Funding Acquisition: B.S.

## Funding

## Competing interests
The authors declare no competing interests.
