## [Transparent Peer Review file · Communications Psychology]

AI outperforms humans in establishing interpersonal closeness in emotionally engaging interactions, but only when labelled as human

Corresponding Author: Dr Tobias Kleinert

Version 0:

Decision Letter:

Dear Dr Kleinert,

Thank you for your patience during the peer-review process. Your manuscript titled "AI outperforms humans in establishing interpersonal closeness in emotionally engaging interactions – but only when labelled as human" has now been seen by 2 reviewers, and I include their comments at the end of this message. They find your work of interest but raised some important points. We are interested in the possibility of publishing your study in Communications Psychology, but would like to consider your responses to these concerns and assess a revised manuscript before we make a final decision on publication.

We therefore invite you to revise and resubmit your manuscript, along with a point-by-point response to the reviewers. Please highlight all changes in the manuscript text file.

Editorially, we consider it crucial that all methodological concerns raised by Reviewer #1 are thoroughly addressed. On the note of preregistration, please ensure your revisions bring the paper into alignment with our preregistration policy <https://www.nature.com/commpsychol/editorial-policies/preregistration-policy>. Authors must disclose all deviations from the preregistered protocol and explain the rationale for deviation (e.g., flaw, feasibility, suboptimality). In cases of deviation from the preregistered analysis plan for reasons other than fundamental flaw or feasibility, the originally planned analyses must also be reported. Exploratory analyses are welcome provided they are labelled as such.

I am attaching an Editorial Requests Table that details critical reporting requirements for the revised manuscript. Please attend to each item and ensure your manuscript is fully compliant. If your revised manuscript is not aligned with these requests on major issues, such as those concerning statistics, it may be returned to you for further revisions without re-review.

Please submit the following items:

- Revised manuscript
- Point-by-point response to the referees' comments
- Cover letter (as a separate document)
- [Nature Research Reporting Summary](https://www.nature.com/documents/nr-reporting-summary.pdf)
- Completed Editorial Request Table (attached).

via this link: Link Redacted .

Additional guidance is available in our style and formatting guide Communications Psychology formatting guide.

Best regards,

Marike

Marike Schiffer, PhD
Chief Editor
Communications Psychology

REVIEWER EXPERTISE:

Reviewer #1 human-AI interaction

Reviewer #2 human-AI interaction

REVIEWER REPORTS:

Reviewer #1 (Remarks to the Author):

Across two studies the authors consider relationship building with AI, both when people do not know they are communicating with AI (Study 1) and when they do (Study 2). I think that the question the authors are focusing on is highly interesting, and that their findings are thought-provoking for both conceptual and practical reasons. That being said, I had a number of concerns about the paper in its current form. I list these below in the hope that they are of use to the authors as they continue their work in this area.

1. Conceptual clarity: The authors situate their research within recent findings on social-emotional interactions with AI and how these compare with such interactions with human partners. As the authors articulate, understanding this question has both basic and applied implications, and it is thus being concurrently investigated by academic researchers and by for-profit companies. The authors bring in the novel application of the Fast Friends procedure, which is potentially quite interesting; that being said, it wasn't clear from the writing why this particular procedure was used, and how it uniquely might help us understand questions that we do not yet have insight into. Also relevant to this point, it was not until I looked at Table S1 in the supplement that I understood that the AI was prompted to respond as a fictional character; this is a key detail that fundamentally impacts the meaning of the findings and, potentially, the way that they should be situated in the literature. That is, to the degree that the mechanism of the current effects is self-disclosure, it makes sense that AI self-discloses when it is 'in character'. But, that is a fairly specific type of AI-human social interaction, which doesn't necessarily fit with some of the broader literature cited to situate the research. Bringing more conceptual clarity to what the authors are critically interested in, and how their choice of procedure uniquely provides insight into that issue, would do a lot to help clarify the current contribution.

2. Methodological choices: A few of the methodological choices the authors made gave me some pause. First, the critical comparison in Study 1 is between AI and human responses. I was a bit surprised that these were operationalized using 6 responses for each category – I appreciate the choice of human responses was randomly made from the set of 20 that the authors collected, but it seems a bit underwhelming to compare AI and human responses with ultimately an N of 6 per condition. (One could have run these studies by actually having participants interact with actual human responders in each case; or with AI acting as a human.) A related minor question I had was why, given that only 6 human responses were used, the AI wasn't given an actual character profile of the human in question. This would have seemed an easy way to at least

equate the human and AI (in character) on basic characteristics.

I was also a bit confused by the design of Study 1 and Study 2, which used some overlapping data and appear to have been collected concurrently. Essentially, this was run as one big study if I understand correctly, with participants randomly assigned to one of 6 conditions (ai source/human label/small talk ; human source/human label/small talk ; ai source/human label/deep talk ; human source/human label/deep talk ; ai source/ai label/deep talk ; human source/ai label/deep talk) – the first four conditions were analyzed as “Study 1” and the last four conditions were analyzed as “Study 2” (with the middle two conditions being used in both cases). Given that the data was all collected at one time, this may be reasonable, but I think it needs to be spelled out a lot more clearly, and the reason for this design choice needs to be articulated.

Another source of confusion is the baseline measurement of closeness, and its differential use across analysis. If the difference from pre to post interaction is considered what is essential (as suggested by the initial analysis), why is the baseline measure not considered in the main analysis? (In regard to the analysis using the baseline measure for Study 1, the authors mention that “the increase in closeness was even marginally lower in interactions with humans compared to interactions with AI (labelled as human) across emotional intensity levels” (line 124-125) – would be great to provide related statistics for this statement.)

Another important point is for the authors to engage with their original hypotheses in some fashion. From the preregistrations, they actually had a quite different set of predictions (notably, they expected humans to outperform AI), and reference to that should be made somewhere. From my perspective, that their original hypotheses were not supported is not a negative; it should just be noted. It would also be good to clarify the text analyses were exploratory in nature, as they are not mentioned in the preregistration (I think the authors do this in the method section, but not the results).

3. Bigger picture implications. This is not at all a critical comment, but my own grappling with the meaning and implications of your findings -- I think in some way the most interesting finding to me is that people do form closeness to AI (from pre to post), even when they know that the responses are coming from AI acting as a character (i.e., the AI says that they are a student studying xyz at a university). In some ways this is genuinely surprising to me, given that AI does not have a self and is therefore acting only “in role” rather than actually disclosing real information – it’s interesting that this doesn’t cross the “uncanny valley” type of experience, and I wonder if the authors have more insight into why this might be. Are people essentially role playing and feeling close to the character that is being enacted? Or are they actually growing in closeness to the AI itself? Do they see the AI as authentic – in that it is expressing the role authentically – or as inauthentic, as it is essentially fabricating responses (an AI doesn’t have dreams or memories)? It might be interesting if the authors collected any relevant individual differences (AI aversion, basic personality, etc.) to explore factors that might moderate this pattern in order to provide more insight into what is driving it.

I wish the authors continued success as they continue to pursue this interesting line of work!

Reviewer #2 (Remarks to the Author):

The authors conducted two rigorous studies to assess whether humans develop interpersonal closeness with an AI partner during the Fast Friends closeness-generation procedure. In my opinion, the studies are well-designed to test this question, and the authors justified how their design decisions enhanced our ability to interpret their results (e.g., avoiding history effects by conducting both studies simultaneously, using AI generated prompts to ensure that partner characteristics did not introduce a confound; having random response times to enhance realism). The use of the fast friends procedure is also a notable strength, as it is a well-established method for developing closeness.

The burgeoning AI literature was also well-reviewed and described. In addition to the many empirical studies they cite, I would encourage the authors to review Machia et al.'s (2024) Social Need Fulfillment Model for Human–AI Relationships, which aims to apply theory to explain whether and when AI partners can meet social needs.

Finally, I commend the authors for thorough methods and results reporting as well as a well-written manuscript that highlights key findings and as a balanced description of potential implications.

* TRANSPARENT PEER REVIEW: Communications Psychology uses a transparent peer review system. This means that we publish the editorial decision letters including Reviewers' comments to the authors and the author rebuttal letters online as a supplementary peer review file. However, on author request, confidential information and data can be removed from the published reviewer reports and rebuttal letters prior to publication. If your manuscript has been previously reviewed at another journal, those Reviewers' comments would not form part of the published peer review file.

Communications Psychology is committed to improving transparency in authorship. As part of our efforts in this direction, we are now requesting that all authors identified as ‘corresponding author’ create and link their Open Researcher and

Contributor Identifier (ORCID) with their account on the Manuscript Tracking System prior to acceptance. ORCID helps the scientific community achieve unambiguous attribution of all scholarly contributions. You can create and link your ORCID from the home page of the Manuscript Tracking System by clicking on 'Modify my Springer Nature account' and following the instructions in the link below. Please also inform all co-authors that they can add their ORCIDs to their accounts and that they must do so prior to acceptance.

Version 1:

Decision Letter:

Dear Dr Kleinert,

Thank you for your patience during the peer-review process. Your manuscript titled "AI outperforms humans in establishing interpersonal closeness in emotionally engaging interactions – but only when labelled as human" has now been seen by the 2 previous reviewers, and I include their comments at the end of this message. Reviewer #2 is satisfied with the revisions, but Reviewer #1 highlights some persisting concerns.

We therefore invite you to revise and resubmit your manuscript, along with a point-by-point response to the reviewers. Please highlight all changes in the manuscript text file.

Editorially, we consider it necessary that the referees' concerns are comprehensively addressed, even if no further data are collected. We also highlight that for preregistered work, we expect the preregistered hypotheses to be listed in the Introduction or Methods section verbatim. Throughout the sections (Methods and Results), the mapping should be clear, i.e. it should be spelt out which hypotheses were preregistered and which analyses exactly inform said hypotheses.

Please also note that null results must be reported in full (not just the p-values) and that non-significant findings in NHST may not be interpreted as evidence for the absence of an effect or a difference (eg. line 494).

Please submit the following items:

- Revised manuscript
- Point-by-point response to the referees' comments
- Cover letter (as a separate document)
- [Nature Research Reporting Summary](https://www.nature.com/documents/nr-reporting-summary.pdf)

via this link: Link Redacted .

Additional guidance is available in our style and formatting guide [Communications Psychology formatting guide](https://www.nature.com/documents/commpsychol-style-formatting-guide-accept.pdf).

We would appreciate it if you could keep us informed about an estimated timescale for resubmission, to facilitate our

planning.

Best regards,

Marike

Marike Schiffer, PhD
Chief Editor
Communications Psychology

REVIEWER REPORTS:

Reviewer #1 (Remarks to the Author):

Thank you for the opportunity to comment on this revised paper. I think that the additions and clarifications made by the authors are useful, and appreciate their efforts to respond to my earlier comments.

To clarify some of my earlier comments, one methodological concern that I had involved the matching of human and AI responses. From the authors' response, I do not think my concern was clear. It seems that the authors told the AI to create six fictional characters and respond from the perspective of those characters. My question was why they did not just provide the real six profiles of the humans who provided the human stimuli (i.e., the humans who participated in the laboratory session and were selected as the human stimuli) to the AI and ask it to respond from the perspective of those identities. This would allow the authors to control at a broad level for the character itself.

A second concern involved the collection of all of the data at the same time. I appreciate the explanation offered by the authors, although I'm not sure I understand it. First, I'm not sure why there is more concern about repeat participants here than in any other multi-study package – there are ways to prevent repeat participants that are routinely used. Regarding history effects, I'm not quite sure what this means. I had assumed the worry was history concerns within the AI prompting – there is a concern in repeatedly asking an AI the same question, although LLMs allows you to clear memory to avoid this problem. More substantively, though, this is not a concern in the current case as AI was only used at one moment in time to generate the stimuli (i.e., it wasn't used in the actual studies). The authors may be referring to some other history effect I am not thinking of; perhaps explaining briefly what that is will be useful.

I appreciated very much the addition of information about how the current results fit the authors' preregistered hypotheses. However, the manuscript now reads a bit oddly, as the hypotheses are not listed anywhere within the manuscript (see lines 95-96 which direct readers to the OSF to see the hypotheses) but then the results section refers back to specific hypotheses; in addition, the writeup itself doesn't really imply what was expected (i.e., from the intro I would not have expected the hypotheses that were preregistered). Thus, when the authors say their findings were surprising, or that specific hypotheses were not supported, it reads a bit jarring. Perhaps a footnote or short paragraph earlier simply explaining the evolution of the authors' thinking would make this easier to follow? I leave guidance on this to the editor.

Finally, just a note that I appreciated the content you added to try to grapple with the meaning of your findings (pg 14) – I think wrestling with the bigger picture implications here is very useful!

Reviewer #2 (Remarks to the Author):

In addition to the one point I raised, Reviewer 1 raised several important points. I think the authors did an excellent job responding to all points raised and clarifying the revised manuscript.

Version 2:

Decision Letter:

Dear Dr Kleinert,

Your manuscript titled "AI outperforms humans in establishing interpersonal closeness in emotionally engaging interactions – but only when labelled as human" has now been editorially evaluated and I am delighted to say that we are happy, in principle, to publish a suitably revised version in Communications Psychology.

We therefore invite you to revise your paper one last time to address the remaining concerns. At the same time we ask that you edit your manuscript to comply with our format requirements and to maximise the accessibility and therefore the impact of your work.

EDITORIAL REQUESTS:

SUBMISSION INFORMATION:

OPEN ACCESS:

*** TRANSPARENT PEER REVIEW:** Communications Psychology uses a transparent peer review system. On author request, confidential information and data can be removed from the published reviewer reports and rebuttal letters prior to publication. If you are concerned about the release of confidential data, please let us know specifically what information you would like to have removed. Please note that we cannot incorporate redactions for any other reasons.

Link Redacted

Best regards,

Marika

Marika Schiffer, PhD
Chief Editor
Communications Psychology

Point-by-point responses to reviewers

Reviewer #1 (Remarks to the Author):

Across two studies the authors consider relationship building with AI, both when people do not know they are communicating with AI (Study 1) and when they do (Study 2). I think that the question the authors are focusing on is highly interesting, and that their findings are thought-provoking for both conceptual and practical reasons. That being said, I had a number of concerns about the paper in its current form. I list these below in the hope that they are of use to the authors as they continue their work in this area.

We sincerely thank Reviewer 1 for their thoughtful and constructive feedback. We are especially grateful for the positive evaluation of our research question as “highly interesting” and our findings as “thought-provoking for both conceptual and practical reasons.” We also appreciate the time and care taken to provide detailed suggestions to strengthen our manuscript. In the following, we address each of the reviewer’s concerns point by point, and provide the corresponding revisions made to the manuscript. Bold text indicates changes made to the manuscript for your convenience. In the revised manuscript, all changes made in response to reviewer comments are highlighted in yellow, and those made to address editorial policies are highlighted in turquoise.

1. Conceptual clarity: The authors situate their research within recent findings on social-emotional interactions with AI and how these compare with such interactions with human partners. As the authors articulate, understanding this question has both basic and applied implications, and it is thus being concurrently investigated by academic researchers and by for-profit companies. The authors bring in the novel application of the Fast Friends procedure, which is potentially quite interesting; that being said, it wasn’t clear from the writing why this particular procedure was used, and how it uniquely might help us understand questions that we do not yet have insight into. Also relevant to this point, it was not until I looked at Table S1 in the supplement that I understood that the AI was prompted to respond as a fictional character; this is a key detail that fundamentally impacts the meaning of the findings and, potentially, the way that they should be situated in the literature. That is, to the degree that the mechanism of the current effects is self-disclosure, it makes sense that AI self-discloses when it is ‘in character’. But, that is a fairly specific type of AI-human social interaction, which doesn’t necessarily fit with some of the broader literature cited to situate the research. Bringing more conceptual clarity to what the authors are critically interested in, and how their choice of procedure uniquely provides insight into that issue, would do a lot to help clarify the current contribution.

Thanks to this comment, we now clarify why the Fast Friends Procedure was chosen as an interaction protocol in the theory section of the manuscript. Please also note that the choice of the FFP was explicitly commended by Reviewer 2 (“The use of the fast friends procedure is also a notable strength, as it is a well-established method for developing closeness”).

LI. 103-105 (theory): **“We specifically selected the FFP because it was designed to enable the rapid development of interpersonal closeness in the early stages of relationship building between previously unacquainted partners (Aron et al., 1997), which was the focus of our study.”**

Another important point raised by the reviewer is that the AI responds as a fictional character to the FFP items, rather than in its “original form.” We now provide an explanation for this choice in the theory section, and discuss its consequences in the limitations subsection of the discussion.

LI. 109-111 (theory): **“We prompted the AI to respond from the perspective of fictional characters rather than in its original form to enable it to answer personal questions and keep basic character information (name, age, place of residence, field of study) consistent with human partners.”**

LI. 666-674 (discussion): **“Additionally, the AI was prompted to respond from the perspective of six students. Although we used only a one-sentence minimal prompt to keep responses as close as possible to standard AI output, this approach may still affect how our findings relate to typical AI interactions, which do not involve such prompting. Importantly, however, the prompt did not include instructions regarding the tone of the interaction (e.g., the amount of self-disclosure, empathy, or emotionality), demonstrating that AI-generated responses showed self-disclosure and fostered relationship building even without specific prompting to do so.”**

Additionally, we now provide a more precise explanation of the study’s specific research focus in the theory:

LI. 43-47 (theory): **“However, it remains an open question whether, and under what conditions, humans build relationships with AI to the same extent as with other humans, especially in the early stages of building a new relationship to a previously unknown other.** The present study aims to fill this research gap by investigating differences in relationship building between **initial** interactions with humans versus AI (i.e., LLM-generated content).”

2. Methodological choices: A few of the methodological choices the authors made gave me some pause. First, the critical comparison in Study 1 is between AI and human responses. I was a bit surprised that these were operationalized using 6 responses for each category – I appreciate the choice of human responses was randomly made from the set of 20 that the authors collected, but it seems a bit underwhelming to compare AI and human responses with ultimately an N of 6 per condition. (One could have run these studies by actually having participants interact with actual human responders in each case; or with AI acting as a human.)

Thank you for raising this point. Thanks to your comment, we now explain the choice of the number of characters in the method section and address this limitation in the discussion section.

LI. 208-210 (methods): **“As we found that the plausibility of AI-generated characters decreased as their number increased, we set six as a compromise between plausibility and representability.”**

LI. 672-674 (discussion): **“Furthermore, including more than six human and AI interaction partners per condition could be beneficial in future studies to better represent typical human and AI responding.**

A related minor question I had was why, given that only 6 human responses were used, the AI wasn’t given an actual character profile of the human in question. This would have seemed an easy way to at least equate the human and AI (in character) on basic characteristics.

Thanks to your comment, we now describe the matching of AI and human character information in more detail.

LI. 216-219 (methods): **“Note that responses to warm-up items for both human and AI interaction partners were drawn from the aforementioned AI-generated characters to maintain consistency across conditions in terms of name, age, place of residence, and field of study.”**

For clarification, we now also provide this explanation in the methods section “Online experiment”:

LI. 245-248 (methods): **“To ensure consistency in basic character information across conditions, participants in both the human and AI interaction groups received AI-generated responses to the warm-up items, while no personal information from human partners was used.”**

I was also a bit confused by the design of Study 1 and Study 2, which used some overlapping data and appear to have been collected concurrently. Essentially, this was run as one big study if I understand correctly, with participants randomly assigned to one of 6 conditions (ai source/human label/small talk ; human source/human label/small talk ; ai source/human label/deep talk ; human source/human label/deep talk ; ai source/ai label/deep talk ; human source/ai label/deep talk) – the first four conditions were analyzed as “Study 1” and the last four conditions were analyzed as “Study 2” (with the middle two conditions being used in both cases). Given that the data was all collected at one time, this may be reasonable, but I think it needs to be spelled out a lot more clearly, and the reason for this design choice needs to be articulated.

Your comment made us aware that the reasons for the simultaneous data collection and the shared data for both studies were not articulated clearly enough in the previous version of the manuscript. The reasons for this choice were to prevent history effects (a point Reviewer 2 explicitly commended), and to prevent participants from taking part in both studies. To increase transparency, we now communicate the design and provide an explanation for this choice already in the theory section:

LI. 96-99 (theory): **“Both studies were conducted online simultaneously and included some shared data to enhance comparability between studies, prevent participants from taking part in both studies, and avoid history effects, which are likely in a rapidly evolving field such as AI interactions.”**

Another source of confusion is the baseline measurement of closeness, and its differential use across analysis. If the difference from pre to post interaction is considered what is essential (as suggested by the initial analysis), why is the baseline measure not considered in the main analysis? (In regard to the analysis using the baseline measure for Study 1, the authors mention that “the increase in closeness was even marginally lower in interactions with humans compared to interactions with AI (labelled as human) across emotional intensity levels” (line 124-125) – would be great to provide related statistics for this statement.)

Thank you for this comment. As specified in our preregistration, we decided in advance to focus on post-interaction measures of closeness instead of difference scores (i.e., post measures minus pre measures of closeness) in our main analyses (hypotheses 2, 3, 4, 6, 7, 8). The “basic analyses” (hypotheses 1 and 5) were included only as a proof of concept. Your comment made us aware that we need to provide a more detailed explanation for this choice.

LI. 282-290 (methods): **“Note that we refrained from using difference scores (post minus pre) in our main analyses, as pre-measures are likely already influenced by the information from the warm-up items of the FFP (i.e., name, age, residence, and field of study), which can bias social perception (e.g., Sidhu & Pexman, 2015; Vedel, 2016). Subtracting these initial impressions could thus reduce the variance of interest in the post measures of closeness. This issue is further amplified in the two “label AI” conditions, where pre-measures are additionally shaped by participants’ expectations of interacting with AI, meaning that differencing would remove precisely those initial attitudes that are central to our research question. Consistent with our preregistration, we therefore relied on post measures only.”**

Furthermore, the reviewer rightfully points out that we provided no statistics for the statement that “the increase in closeness was even marginally lower in interactions with

humans compared to interactions with AI (labelled as human) across emotional intensity levels". As this statement was based merely on a descriptive observation but not on statistical evidence, we have deleted it from the manuscript together with the respective interpretation in the discussion.

Another important point is for the authors to engage with their original hypotheses in some fashion. From the preregistrations, they actually had a quite different set of predictions (notably, they expected humans to outperform AI), and reference to that should be made somewhere. From my perspective, that their original hypotheses were not supported is not a negative; it should just be noted. It would also be good to clarify the text analyses were exploratory in nature, as they are not mentioned in the preregistration (I think the authors do this in the method section, but not the results).

As requested, we now present the results in closer alignment with the preregistered hypotheses and communicate more explicitly the exploratory nature of the language analyses.

Changes regarding the alignment with pre-registered hypotheses:

LI. 371-373 (results): **"As predicted, deep-talk interactions with AI significantly increased interpersonal closeness compared to a baseline measure, confirming hypothesis 1."**

LI. 392-397 (results): **"Surprisingly, these results demonstrate that interactions with humans did not yield stronger feelings of closeness than interactions with AI, and that deep-talk interactions did not yield stronger feelings of closeness than small-talk interactions, leading to the rejection of hypotheses 2 and 3. Furthermore, and contrary to hypothesis 4, AI interactions yielded stronger feelings of closeness than human interactions, but only in the deep-talk condition, not in the small-talk condition."**

LI. 466-468 (results): "These findings demonstrate that relationship building occurred even when participants were informed they were interacting with an AI, **supporting hypothesis 5.**"

LI. 482-488 (results): **"These results show once more that participants did not establish stronger feelings of closeness with human interaction partners than with AI interaction partners. Thus, hypothesis 6 is rejected. As expected, labelling the interaction partner as an AI led to lower interpersonal closeness ratings after the interaction compared to when the partner was labelled as human, demonstrating an anti-AI bias and confirming hypothesis 7. Hypothesis 8 was rejected, as there were no differences in interpersonal closeness (human > AI) following human interactions compared to AI interactions."**

Changes regarding the clarification of the exploratory nature of the language analyses in the results section:

LI. 113-114 (theory): "Furthermore, we applied **exploratory** automated linguistic analysis using the Linguistic Inquiry and Word Count system [...]"

LI. 398-401 (results): "To further **explore** why people feel closer to the AI than to humans following deep-talk interactions, we tested whether AI-generated responses differed from human-generated responses regarding self-disclosure as measured by the Linguistic Inquiry and Word Count system (LIWC-22; Boyd et al., 2022; Tausczik & Pennebaker, 2010)."

LI. 491-493 (results): "We therefore tested whether people themselves communicated differently with AI-labelled partners than human-labelled partners **in an exploratory fashion.**"

3. Bigger picture implications. This is not at all a critical comment, but my own grappling with the meaning and implications of your findings -- I think in some way the most interesting finding to me is that people do form closeness to AI (from pre to post), even when they know that the responses are

coming from AI acting as a character (i.e., the AI says that they are a student studying xyz at a university). In some ways this is genuinely surprising to me, given that AI does not have a self and is therefore acting only “in role” rather than actually disclosing real information – it’s interesting that this doesn’t cross the “uncanny valley” type of experience, and I wonder if the authors have more insight into why this might be. Are people essentially role playing and feeling close to the character that is being enacted? Or are they actually growing in closeness to the AI itself? Do they see the AI as authentic – in that it is expressing the role authentically – or as inauthentic, as it is essentially fabricating responses (an AI doesn’t have dreams or memories)? It might be interesting if the authors collected any relevant individual differences (AI aversion, basic personality, etc.) to explore factors that might moderate this pattern in order to provide more insight into what is driving it.

Thank you for this thought-provoking comment. To the best of our knowledge, the uncanny valley effect mainly refers to visual appearance, movements, or speech in artificial agents, and not, or at least to a lesser degree, to text-based interactions as used in our study. Nevertheless, we fully agree that the finding that people build relationships with AI similarly to humans is highly interesting and was underemphasized in the discussion. In particular, it raises the question about why some people form social relationships with AI while others stay reluctant, and whether other characteristics drive these differences. Accordingly, we added an additional exploratory analysis to the results section suggesting that the anti-AI bias is moderated by personal values associated with natural human communication (i.e., universalism). Furthermore, we now provide a more detailed interpretation of the aforementioned topics in the discussion.

LI. 518-528 (results): **“We found evidence that individuals form social bonds with AI even when being aware of interacting with an artificial agent, yet also observed an anti-AI bias leading to lower feelings of closeness after the interaction compared to human-labelled interactions. To explore why some people form social bonds to AI while others do not, we examined whether AI scepticism is more pronounced in individuals who value natural human communication. Indeed, we found a significant interaction effect between ‘source label’ and universalism, a personal value centred on concern for the welfare of people and nature (Schwartz, 2012), in predicting interpersonal closeness ($F_{(1, 328)} = 4.11, p = .043, \eta^2_p = .012$). This effect was driven by a positive association between universalism and closeness in human-labelled interactions ($\beta_{(162)} = .203, p = .009$), which was not present in AI-labelled interactions ($\beta_{(162)} = .011, p = .891$). These findings indicate that personal values modulate relationship building with AI.”**

LI. 593-607 (discussion): **“So why do individuals form relationships with AI at all, even when being aware of its artificial nature (also see Chaturvedi et al., 2023)? One explanation is the phenomenon of *anthropomorphism*, the tendency to attribute human traits, emotions, or intentions to non-human entities (e.g., Nass & Moon, 2000; Reeves & Nass, 1996). Humans are inherently social, so when presented with AI-generated cues that resemble human-generated cues, the brain may intuitively respond to these artificial cues much as it would to genuine social cues. At the same time, our results highlight that at least some individuals remain reluctant to engage with AI, leading to the question of how these people differ from those who are more receptive. In an exploratory analysis, we found that universalism moderated the difference in interpersonal closeness following AI- versus human-labelled interactions. Specifically, individuals high in universalism felt closer to humans, but not to AI. This indicates that traits linked to natural social interaction, and, vice versa, potentially to negative attitudes towards artificial interaction, may reduce the likelihood of forming bonds with AI. However, this result requires further validation in future research.**

I wish the authors continued success as they continue to pursue this interesting line of work!

Thank you very much!

Reviewer #2 (Remarks to the Author):

The authors conducted two rigorous studies to assess whether humans develop interpersonal closeness with an AI partner during the Fast Friends closeness-generation procedure. In my opinion, the studies are well-designed to test this question, and the authors justified how their design decisions enhanced our ability to interpret their results (e.g., avoiding history effects by conducting both studies simultaneously, using AI generated prompts to ensure that partner characteristics did not introduce a confound; having random response times to enhance realism). The use of the fast friends procedure is also a notable strength, as it is a well-established method for developing closeness.

The burgeoning AI literature was also well-reviewed and described. In addition to the many empirical studies they cite, I would encourage the authors to review Machia et al.'s (2024) Social Need Fulfillment Model for Human–AI Relationships, which aims to apply theory to explain whether and when AI partners can meet social needs.

Finally, I commend the authors for thorough methods and results reporting as well as a well-written manuscript that highlights key findings and as a balanced description of potential implications.

We sincerely thank Reviewer 2 for their thoughtful and encouraging review. We are grateful for your positive evaluation of our study design, methodological decisions, and clarity in reporting. We also thank you for pointing us to Machia et al.'s (2024) Social Need Fulfillment Model for Human–AI Relationships. We have now incorporated this model in the discussion of the revised manuscript:

LI. 585-588 (discussion): “These findings align with the Social Need Fulfillment Model for Human-AI relationships (Machia et al., 2024), which suggests that human-AI interactions typically satisfy only concrete social needs (e.g., pleasure) rather than deeper, symbolic needs (e.g., feeling genuine care).”

Point-by-point responses to reviewers

Reviewer #1 (Remarks to the Author):

Thank you for the opportunity to comment on this revised paper. I think that the additions and clarifications made by the authors are useful, and appreciate their efforts to respond to my earlier comments.

Thank you once again for your constructive feedback. We appreciate your recognition of our efforts to improve the manuscript based on your earlier comments. Below, we provide a detailed, point-by-point response to your remaining points along with the corresponding revisions made to the manuscript.

To clarify some of my earlier comments, one methodological concern that I had involved the matching of human and AI responses. From the authors' response, I do not think my concern was clear. It seems that the authors told the AI to create six fictional characters and respond from the perspective of those characters. My question was why they did not just provide the real six profiles of the humans who provided the human stimuli (i.e., the humans who participated in the laboratory session and were selected as the human stimuli) to the AI and ask it to respond from the perspective of those identities. This would allow the authors to control at a broad level for the character itself.

We appreciate the reviewer's clarification of this point, which we now understand more clearly. We have added a note to the limitations section to acknowledge this issue and highlight the advantages of the reviewer's proposed alternative approach.

LI. 734-739 (discussion): ***"As noted by a reviewer, an alternative and less minimal prompt could be to instruct the AI to respond from the perspective of the specific human profiles used in this study, which could provide broader control for the characters presented. Relatedly, including more than six human and AI interaction partners per condition could be beneficial in future studies to better represent typical human and AI responding."***

Furthermore, we have slightly adapted the explanation of the rationale behind our AI prompt in the methods section:

LI. 257-258 (methods): ***"The following minimal prompt was used to keep AI responses as close as possible to its default style"*** (instead of: "The following minimal prompt was used")

A second concern involved the collection of all of the data at the same time. I appreciate the explanation offered by the authors, although I'm not sure I understand it. First, I'm not sure why there is more concern about repeat participants here than in any other multi-study package – there are ways to prevent repeat participants that are routinely used. Regarding history effects, I'm not quite sure what this means. I had assumed the worry was history concerns within the AI prompting – there is a concern in repeatedly asking an AI the same question, although LLMs allows you to clear memory to avoid this problem. More substantively, though, this is not a concern in the current case as AI was only used at one moment in time to generate the stimuli (i.e., it wasn't used in the actual studies). The authors may be referring to some other history effect I am not thinking of; perhaps explaining briefly what that is will be useful.

Thank you for this comment. We agree that there are effective technical measures to prevent participants from taking part in related studies (e.g., restricting participation based on identical email addresses or IPs). However, these methods can be circumvented (e.g., by changing these identifiers), as we have experienced repeatedly in previous studies. More importantly, our main reason for conducting simultaneous data collection was to minimise

potential history effects. We appreciate the reviewer's comment, which made us realise that a more detailed explanation of this rationale was warranted. We have now added this clarification to the introduction of the revised manuscript.

LI. 99-102 (introduction): ***"History effects are important to consider, as participants' attitudes towards, and interactions with AI can change significantly over time (e.g., Modhvadia et al., 2025), potentially leading to observed differences in AI interactions that reflect broader societal shifts rather than effects of experimental conditions."***

I appreciated very much the addition of information about how the current results fit the authors' preregistered hypotheses. However, the manuscript now reads a bit oddly, as the hypotheses are not listed anywhere within the manuscript (see lines 95-96 which direct readers to the OSF to see the hypotheses) but then the results section refers back to specific hypotheses; in addition, the writeup itself doesn't really imply what was expected (i.e., from the intro I would not have expected the hypotheses that were preregistered). Thus, when the authors say their findings were surprising, or that specific hypotheses were not supported, it reads a bit jarring. Perhaps a footnote or short paragraph earlier simply explaining the evolution of the authors' thinking would make this easier to follow? I leave guidance on this to the editor.

We thank the reviewer for this comment and the editor for guiding us in how to respond to it. We have now added the hypotheses 'verbatim' at the end of the Introduction to improve clarity and alignment between the pre-registration and the current manuscript. Please note that we made minor adjustments to account for updated factor labels as used in the current study (i.e., 'source identity' instead of 'partner identity'; 'emotional intensity' instead of 'depth of interaction'; 'source label' instead of 'information on the identity of the interaction partner'; 'label human' and 'label AI' instead of 'information: human' and 'information: AI'). Furthermore, we focused on interpersonal closeness as the dependent variable, and did not include hypotheses related trust, to avoid confusion (as trust was not further considered in the paper due to the already described extreme ceiling effects). As suggested, we also added brief explanations of how we derived these hypotheses, complementing the existing information to make the rationale behind our predictions more transparent to the reader.

LI. 125-168 (introduction): ***"In summary, the hypotheses we tested in study 1 were: Hypothesis 1: We expect that deep talk interactions with an AI will lead to a significant increase in interpersonal closeness compared to a baseline measure. This hypothesis is being tested to validate findings suggesting that relationship-building with AI is possible (e.g., Yin et al., 2024). We additionally examine whether small-talk interactions will also generate increases in closeness. Hypothesis 2: We expect higher interpersonal closeness towards the interaction partner after interactions with humans compared to interactions with an AI across both small-talk and deep-talk interactions (main effect of the factor 'source identity'). This assumption builds on the traditional view that interactions with humans should elicit greater closeness than interactions with AI, as social interactions are fundamentally rooted in human behaviour (e.g., Wu, 2024). Hypothesis 3: We expect higher interpersonal closeness towards the interaction partner after deep-talk interactions compared to small-talk interactions across interactions with humans and AI (main effect of the factor 'emotional intensity'). This hypothesis relies on the idea that self-disclosure on emotional topics is a key driver of early relationship-building (Aron et al., 1997). Hypothesis 4: We expect that the differences (human > AI) regarding interpersonal closeness towards the interaction partner are larger after deep talk interactions compared to small talk interactions (interaction effect between the factors 'emotional intensity' and 'source identity'). This hypothesis assumes that AI may adequately mimic non-personal small-talk, but not personal deep-talk, as emotions are widely considered a uniquely human domain (e.g., Martinez-Miranda & Aldea, 2005), although more recent research (some published after our pre-registration) suggests otherwise (e.g., Ovsyannikova et al., 2025)."***

The following hypotheses were tested in study 2: Hypothesis 5: We expect that deep talk interactions with an AI will lead to a significant increase in interpersonal closeness compared to a baseline measure, even if participants are informed that they will interact with an AI. Although research indicates that people often hold reservations about AI interactions (e.g., Yin et al., 2024), we anticipate that participants will still develop some degree of perceived closeness with the AI, reflecting the human tendency to respond to human-like artificial agents as if they were real social partners, a phenomenon known as anthropomorphism (e.g., Nass & Moon, 2000; Reeves & Nass, 1996). Hypothesis 6: Analogous to study 1, we expect higher interpersonal closeness towards the interaction partner after interactions with humans compared to interactions with an AI across both 'label human' and 'label AI' interactions (main effect of the factor 'source identity'). Hypothesis 7: We expect higher interpersonal closeness towards the interaction partner when participants are informed that they interact with a human compared to when participants are informed that they interact with an AI across actual interactions with humans and AI (main effect of the factor 'source label'). This hypothesis is based on findings indicating reservations towards social interactions with AI (e.g., Yin et al., 2024). Hypothesis 8: We expect that the differences (human > AI) regarding interpersonal closeness towards the interaction partner are larger when participants are informed that they are interacting with a human compared to when they are informed that they are interacting with AI (interaction effect between the factors 'source identity' and 'source label'). This hypothesis draws on the assumption that the anti-AI bias in the AI-labelled condition would reduce feelings of closeness regardless of the actual source identity, whereas in the human-labelled condition, the expected advantage of genuine human responses would be more apparent."

Finally, just a note that I appreciated the content you added to try to grapple with the meaning of your findings (pg 14) – I think wrestling with the bigger picture implications here is very useful!

We thank the reviewer for their positive feedback and appreciate their recognition of our efforts to engage with the broader implications of our findings.

Reviewer #2 (Remarks to the Author):

In addition to the one point I raised, Reviewer 1 raised several important points. I think the authors did an excellent job responding to all points raised and clarifying the revised manuscript.

We thank the reviewer once again for their constructive feedback.

Editorial requests:

Editorially, we consider it necessary that the referees' concerns are comprehensively addressed, even if no further data are collected. We also highlight that for preregistered work, we expect the preregistered hypotheses to be listed in the Introduction or Methods section verbatim. Throughout the sections (Methods and Results), the mapping should be clear, i.e. it should be spelt out which hypotheses were preregistered and which analyses exactly inform said hypotheses.

Thank you for this feedback. We have added the pre-registered hypotheses 'verbatim' in the introduction (for details, see response to reviewer 1's third comment). The hypotheses are clearly mapped to specific analyses in the statistical analyses section in the methods and to specific results in the results section. All hypotheses were pre-registered.

Please also note that null results must be reported in full (not just the p -values) and that non-significant findings in NHST may not be interpreted as evidence for the absence of an effect or a difference (eg. line 494).

As requested, we now provide full reports of null results across the results section.

Ll. 461-463 (results): ***“[...] no significant difference regarding response length; $F(1, 10) = 18.57, p = .392, \eta^2p = .074, 95\% CI [.000, .407]; MAI = 313.83, SDAI = 18.89; M_{human} = 295.50, SD_{human} = 46.45$ ”***

Ll. 472-474 (results): ***“[...] no significant difference regarding response length; $F(1, 160) = .007, p = .932, \eta^2p < .001, 95\% CI [0.00, .012]; MAI = 197.62, SDAI = 62.58; M_{human} = 199.15, SD_{human} = 64.58$.”***

Ll. 552-553 (results): ***$(F(1, 330) = .050, p = .823, \eta^2p < .001, 95\% CI [.000, .012]; MAI = 34.14, SDAI = 4.39; M_{human} = 34.05, SD_{human} = 4.86)$***

Furthermore, we have added an equivalence test, followed by a Bayesian t -test to our statistical analysis to support our conclusion that “people form *relationships with AI to a similar extent as with other humans when the partner is labeled as human*”:

Ll. 381-385 (methods): ***“To test whether increases in closeness differed meaningfully between human and AI interactions, we conducted equivalence testing (TOST) with bounds of $\pm .35$ pooled standard deviations, representing small-to medium effect sizes. This analysis was complemented by a Bayesian two-sample t -test to quantify evidence for or against a meaningful difference.”***

Ll. 430-434 (results): ***“An equivalence test followed by a Bayes t -test revealed that differences in closeness increases between human and AI interactions across levels of emotional intensity were practically negligible when testing against the presence of a small-to-medium effect ($t_{(139.58)} = -2.71, p = .004$). The Bayes-Factor ($BF_{01} = 7.43$) indicated that the data were 7.43 times more likely to occur under the null hypothesis than under the alternative hypothesis.”***

Finally, we made minor changes to adjust the American spelling to British spelling throughout the paper.